# Prevalence of periodontitis in adolescents: A systematic review protocol

Ana Beatriz de Oliveira Monteiro[1], Lia Rosana Honnef[1],
Júlia Meller Dias de Oliveira[1‡], Patrícia Pauletto[2*¤‡], Carla Massignan[3,4‡],
Cristine Miron Stefani[3‡], Glaucia Santos Zimmermann[5‡], Graziela De Luca Canto[1,6‡]

1 Department of Dentistry, Brazilian Centre for Evidence-Based Research (COBE), Federal University of Santa Catarina (UFSC), Florianópolis, Brazil, 2 School of Dentistry, Universidad De Las Américas (UDLA), Quito, Ecuador, 3 University of Brasilia (UNB), Brasília, Brazil, 4 Department of Pediatric Dentistry, University of Illinois Chicago, Chicago, Illinois, United States of America, 5 Department of Dentistry, Federal University of Santa Catarina (UFSC), Florianópolis, Brazil, 6 Emergency Medicine and Evidence-Based Medicine, Federal University of São Paulo, São Paulo, Brazil

☯ These authors contributed equally to this work.
‡ These authors also contributed equally to this work.
¤ Current address: School of Dentistry, Universidad De Las Américas (UDLA), Quito, Ecuador
* patricia.pauletto@udla.edu.ec

## Abstract

Periodontitis is considered an inflammatory disease that affects the tooth's supporting structures, resulting in its progressive destruction, which can lead to several problems, such as loss of teeth, bone reabsorption, and even exacerbation of chronic systemic diseases. Periodontitis is a common periodontal condition in the adult dentate population, affecting approximately 6 in 10 individuals. Among those affected, approximately 25% develop the most severe form of the disease. The prevalence of periodontitis in adolescents is not well established. Therefore, this study protocol of a systematic review that aims to answer the following research question: "What is the prevalence of periodontitis in adolescents?". This systematic review will include observational studies reporting the prevalence of periodontitis in individuals aged from 13 to 18 years without comorbidities. A search will be conducted in the following databases: EMBASE, LILACS (via BVS), MEDLINE (via PubMed), and Web of Science. The gray literature will be consulted through Google Scholar and ProQuest. No language or time of publication restrictions will be applied. To determine which studies will be included, two independent reviewers will select the articles through a two-phase process. In the first phase, reviewers will assess the titles and abstracts, and, in the second phase, they will read the full texts using predefined eligibility criteria. Data collected will include author, study design, year of publication, country, sample size, sex, age, diagnostic methods, definition of periodontitis, reliability of criteria, examination protocol, case classification and prevalence rates. The risk of bias will be assessed according to the JBI Critical Appraisal Checklist for Studies Reporting Prevalence Data. Results will be synthesized into a narrative description and, if

**Data availability statement:** This article is a systematic review protocol. We intend to report the full results when the research is completed in a new publication.

**Funding:** The author(s) received no specific funding for this work.

**Competing interests:** The authors have declared that no competing interests exist.

**Abbreviations:** AAP, American Academy of Periodontology; BVS, Biblioteca Virtual em Saúde; CAL, Clinical Attachment Loss; CDC, Centers for Disease Control; EFP, European Federation of Periodontology; LILACS, Literatura Latinoamericana y del Caribe en Ciencias de la Salud; MEDLINE, Medical Literature Analysis and Retrieval System Online; OSF, Open Science Framework; POS, Acronym - P: Population, O: Outcomes, S: study design; PPD, Periodontal Pocket Depth; PRISMA-P, Preferred Reporting Items for Systematic Reviews and Meta-Analyses Protocols; PROSPERO, International prospective register of systematic reviews; SR, Systematic Review

possible, a meta-analysis will be conducted. The systematic review methodology will enable a comprehensive and accurate analysis of the available evidence. Moreover, this approach may contribute to clinical decision-making, promoting the use of more appropriate and preventive treatments.

## Background

Periodontitis is characterized as a chronic inflammatory disease associated with dental biofilm dysbiosis, resulting in the progressive destruction of dental support structures [1]. According to the World Workshop on the Classification of Periodontal and Peri-Implant Diseases and Conditions (2017), periodontitis can be classified based on its stage, which is determined by severity (stages I to IV), reflecting biological aspects such as risk of rapid progression, treatment response, and impact on systemic health. Additionally, it can be classified by complexity, extent, and distribution [1–2]. Periodontitis has been studied for years to improve the oral health of the population. Prevalence and incidence are used to measure the frequency of periodontitis, while indices or classifications are employed to quantify the extent and distribution of the disease [3].

A study that collected data from 1990 to 2019, from the Global Burden of Disease 2019 study, reported that periodontitis was negatively associated with the level of socioeconomic development. Periodontitis was more frequent among those aged 55–59 years, but the incidence of periodontitis has shown an increasing trend among younger individuals [4].

Given the evolving prevalence rate in adults, understanding the importance of estimating periodontal disease globally in adolescents is crucial for developing prevention and treatment strategies for this condition. Moreover, it is estimated that by 2050, more than 1.5 billion people will be affected by severe cases of periodontitis, with a significant portion of this population being edentulous. This scenario underscores the urgent need for public policies focused on the prevention and treatment of the disease, as periodontitis increasingly constitutes a pressing global public health issue [5].

A study conducted in Mangalore, India, indicated that the prevalence of periodontitis among adolescents is low, approximately 1.86%, with 1.5% being chronic cases and 0.36% being aggressive cases [6]. By the other hand, another recent study in the south of Thailand, found that early-onset periodontitis in adolescents is very common, with a prevalence of 44.5% [7]. An ecological study on the prevalence of periodontitis in adolescents revealed significant rates. In Norway, the rate reaches 66%, while in Iran it is 30% and in Belarus, 15%. In Germany and Taiwan, 14% of young people were affected by the condition [8]. In light of these data, the importance of early diagnosis becomes evident, considering that periodontitis is a destructive disease. It is essential to make patients, and guardians aware that it is a disease that chronically affects the general population, potentially causing discomfort in the gingival region, bone damage, and premature tooth loss. Furthermore, assisting dentists in the accurate diagnosis of the disease is crucial for developing appropriate treatments

and promoting periodontal health. Moreover, the evidence presents inconsistent data regarding the prevalence and associated risk factors of periodontitis, indicating the need for a systematic review (SR) to consolidate information and provide precise estimates. Understanding the prevalence of periodontitis in different populations is essential to guide prevention and treatment strategies, particularly in vulnerable groups, thereby helping to reduce the progression of the disease into adulthood. Given its global relevance and the regional and socioeconomic variations in the impact of periodontitis among adolescents, prevalence studies provide valuable data to support public health strategies [9]. A previous SR published in 2019 investigated the prevalence of the disease in young individuals, but presented significant methodological limitations, such as restricting studies to those published between 1999 and 2017, and including case reports and case series [10]. To address these limitations, the present SR aims to estimate the prevalence of periodontitis in adolescents aged 13–18 years without restrictions regarding publication year, language, or country, focusing exclusively on cross-sectional studies. This study seeks to provide a precise and comprehensive estimate of this condition within this age group.

## Methods

### Study registration

This SR protocol was developed following the guidelines of Preferred Reporting Items for Systematic Review and Meta-Analyses Protocols (PRISMA-P) (Appendix 1), and it was registered in the International Prospective Register of Systematic Reviews (PROSPERO) [11] on July 17, 2024, database under number CRD42024566374. The protocol details eligibility criteria, selected databases, data collection, and the risk of bias assessment to ensure that this SR will be conducted impartially, transparently, and reproducibly.

### Research question

This SR will answer the focused question: "What is the prevalence of periodontitis in adolescents?"
The question was developed using the adapted POS acronym (Table 1).

### Eligibility criteria

**Inclusion criteria.** Included studies must fit the following criteria:

- Studies including adolescents aged 13–18, regardless of sex.

- Periodontitis should be diagnosed through clinical examination (alone or in combination with radiographic assessment), using clearly defined criteria for case definition, taking into account the clinical characteristics and diagnostic criteria applied in the studies, in addition to the formal definitions established.

- Studies presenting data on the prevalence, with convenience and/or population-based samples.

- Cross-sectional studies.

**Exclusion criteria.** The following studies will be excluded:

1) Studies that included only samples from children, adults or elderly individuals or studies that did not present separate data for adolescents with periodontitis from other age groups.

**Table 1. Research question based on POS acronym.**

| P = Participants | Adolescents, from 13 to 18 years old, regardless the sex. |
|---|---|
| O = Outcomes | Prevalence of periodontitis clinically diagnosed (alone or combined with radiographic assessment), with explicit criteria for case definition. |
| S = Study Design | Cross-sectional studies. |

2) Studies in which the sample is exclusively composed of specific groups, such as people with congenital anomalies, patients with COVID-19 or other systemic diseases.

3) Studies that did not mention the criteria/methods for detection or diagnosis of periodontitis or did not diagnose periodontitis by clinical examination and used only radiography to detect the disease.

4) Studies that did not use a cross-sectional design.

5) Reviews, letters, books, conference abstracts and guidelines.

6) The full text is not available, even after trying to contact the corresponding authors (three attempts in a three-week period).

## Information sources and search strategy

The preliminary search strategy was developed and tested on PubMed on January 4, 2024. After the tests, with the assistance of a health sciences librarian and information specialists the final search strategy was developed for each database. The following databases will be searched EMBASE, LILACS (in Spanish: *Literatura Latinoamericana y del Caribe en Ciencias de la Salud*), MEDLINE (via PubMed) and Web of Science. Gray literature search will be conducted through Google Scholar (100 first hits) and ProQuest Dissertation & Thesis. Additional literature will also be obtained through reference lists of the included studies and expert consultation. The authors with more than four included studies in our SR will be considered experts. The first author (A.B.O.M) will apply the search strategy in each database.

To ensure transparency and integrity in the methodology of this SR protocol, any changes made will be documented at the end of the manuscript. This will encompass adjustments to the inclusion and exclusion criteria, along with the justification for each change and an assessment of its impact on the review's results. We will use the Open Science Framework (OSF) platform as a repository to make the original data available, ensuring transparency and accessibility.

## Study records

**Data management.** The file from each database will be imported by the first author into a reference manager (EndNote-Web; Clarivate, Saint Helier, Jersey), where duplicates will be removed. Subsequently, the references will be imported into a screening software (Rayyan, Qatar Computing Research Institute, Data Analytics, Doha, Qatar).

## Study selection

In the screening software, two independent reviewer's authors (A.B.O.M. and L.R.H.) will select the studies in two phases. In phase 1, they will evaluate the titles and abstracts according to the eligibility criteria. After that, in phase 2, they will analyze the full-texts and select articles by the same criteria. Also, they will crosscheck all the results found. If disagreements arise, a third reviewer (J.M.D.O.) will make a final decision on both phases. If important data for the review are missing or unclear, an attempt will be made to contact the study's corresponding author. The excluded studies, along with the reasons for exclusion, will be presented in Appendix 2 of the final review.

## Data collection and data items

One reviewer (A.B.O.M.) will extract data from the included studies, while a second reviewer (L.R.H.) will verify all extracted information by comparing the data extracted with the original data in the included studies. The data will be stored in a data extraction form that will be developed using a spreadsheet program (Microsoft Excel; Microsoft Corporation, Redmond, WA, USA). If disagreements arise, a third reviewer (J.M.D.O.) will be contacted for a final decision. The collected information will include the author, study design, publication year country, patient characteristics (sample size,

sex, and age), clinical characteristics (diagnostic methods, definition of periodontitis cases, case of periodontitis by sex, periodontal criteria confidence, periodontal examination protocol, periodontitis/healthy or gingivitis and extension/distribution), and outcomes (prevalence rates).

## Risk of bias in individual studies

The risk of bias will be assessed using the JBI Critical Appraisal Checklist for Studies Reporting Prevalence Data (2020) [12] by two independent authors (A.B.O.M. and L.R.H.). After judging all items on the checklist, a general subjective assessment of the answers attributed to each study will be made. To this end, a prior discussion will be conducted among the reviewers to determine the "key" items, i.e., those most critical for assessing the study's risk of bias. Studies that present more positive responses to these "key" items will be classified as having a low risk of bias, while those with negative or uncertain responses will be classified as having a high risk of bias. Therefore, for this prevalence SR, the "key" items were related to the sampling process and sample size (items 1, 2 and 3), and the reliability of the diagnostic methods used (items 6 and 7). In the absence of consensus among the reviewers (A.B.O.M and L.R.H.), the final decision regarding the risk classification of the studies, whether low or high, will be made by a third reviewer (J.M.D.O) [13]. Figures of risk of bias will be created on the *robvis* website (https://www.riskofbias.info/welcome/robvis-visualization-tool) [14].

## Data synthesis

The results will be presented through narrative descriptions and tables containing the characteristics of the clinical studies. If possible, a figure illustrating the world map will be included, highlighting the highest prevalence rates of the disease by country or continent.

If quantitative synthesis is appropriate, a proportion and a subgroup meta-analysis will be performed using the statistical computing software (R Statistics version 3.4.4; The R Foundation for Statistical Computing, Vienna, Austria) using the random effect measure and considering the range 95% confidence level from each study was included [15].

If possible, a meta-regression will be performed to explore the effects of adjustments, particularly on confounding variables, prevalence estimates, female/male ratio, and sample size.

## Heterogeneity

Heterogeneity will be assessed using the visual analysis of the confidence intervals, the Chi$^2$ test ($\chi$2 or chi-squared), Cochran's Q and the $I^2$ statistic. Random effects models will be used for meta-analysis [16].

## Analysis of subgroups

The results will be analyzed in different subgroups based on sex, continent, by categorized case definition and extent (localized or generalized). Meta-analyses are planned for these different groups.

Considering the notable importance of case definitions in prevalence estimates [17], they will be characterized into groups following the strategy employed by Muñoz Aguilera et al. adapted [18] as reliable and unreliable, as described below.

**Reliable case definition of periodontitis.** The following case definitions will be considered reliable:

• Interdental clinical attachment loss (CAL) in ≥ 2 non-adjacent teeth, or buccal or oral CAL ≥ 3 mm with periodontal pocket depth (PPD) >3 mm detectable at ≥ 2 teeth (American Academy of Periodontology [AAP]/European Federation of Periodontology [EFP]) [19].

• Two or more inter-proximal sites with CAL ≥ 3 mm and two or more inter-proximal sites with PPD ≥ 4 mm (not on the same tooth) or one site with PPD ≥ 5 mm (Centers for Diseases Control [CDC]/ AAP, 2012) [20].

- At least two sites on different teeth with clinical attachment level (CAL) 6 mm and at least one site with PPD 4 mm (CDC/AAP 2007) [21].

- Juvenile periodontitis and prepubertal periodontitis: localized and generalized (according to the 1989 classification of the American Academy of Periodontology) [22].

- Generalized chronic periodontitis: at least 30% sites with CAL ≥ 4 mm (CDC 1999) [23].

**Unreliable case definition of periodontitis.** The following reported criteria will be considered as an unreliable case definition: scores on indices such as the Community Periodontal Treatment Needs Index and the World Health Organization Community Periodontal Index 3/4 in at least one quadrant; at least one site with PPD > 4 mm; CAL ≥ 1 mm; Periodontal Attachment Loss Index where the cement-enamel junction to the bottom of the gingival sulcus or periodontal pocket is measured, and the attachment measure is equal to the probing depths and Periodontal Screening and Recording [24].

### Sensitivity analysis

A sensitivity analysis removing studies with a high risk of bias from the meta-analysis and analyzing the impact of the decision to include them is planned.

### Reporting bias assessment

To avoid potential publication bias, a comprehensive literature search will be conducted. To ensure the reliability and impartiality of the research, the affiliations of the sponsors related to the included studies will be analyzed, and any conflicts of interest among the authors will be investigated. If more than ten studies are included, any publication bias will be assessed using the funnel plot method [25].

### Confidence in cumulative evidence

As the GRADE Working Group does not provide guidance for SRs of prevalence and the "Requirements for claiming the use of GRADE" advise against modifying the GRADE approach [26], the certainty of evidence will not be assessed.

## Discussion

As a chronic inflammatory condition traditionally associated with adults, evidence suggests that adolescents, particularly those in at-risk groups, may also be affected, making this a relevant topic for public health and dentistry due to its impact on oral and systemic health. Adolescence is a critical period for establishing healthy habits, and untreated periodontitis during this phase can lead to early tooth loss and systemic complications. However, the literature shows variations in prevalence data and risk factors, highlighting the need for a systematic review to consolidate existing knowledge and provide accurate estimates of the prevalence of this condition among adolescents. Such a review is essential to guide dentists, specialists, and the academic community in the early diagnosis, treatment, and prevention of future periodontal complications, as well as to inform effective prevention strategies.

The systematic review methodology will enable a comprehensive and accurate analysis of the available evidence. Moreover, this approach may contribute to clinical decision-making, promoting the use of more appropriate and preventive treatments.

## Supporting information

**S1 Appendix. PRISMA-P Checklist.**
(DOCX)

**S2 Appendix. Search Strategies.**
(DOCX)

## Acknowledgments

The authors thank the librarian Karyn Lehmkuhl, for her support in developing the search strategy (Federal University of Santa Catarina, Brazil, Email: karyn.lehmkuhl@ufsc.br).

## Author contributions

**Conceptualization:** Ana Beatriz de Oliveira Monteiro, Lia Rosana Honnef, Júlia Meller Dias de Oliveira, Patrícia Pauletto, Carla Massignan, Cristine Miron Stefani, Glaucia Santos Zimmermann, Graziela De Luca Canto.

**Data curation:** Ana Beatriz de Oliveira Monteiro, Lia Rosana Honnef, Júlia Meller Dias de Oliveira, Patrícia Pauletto, Graziela De Luca Canto.

**Formal analysis:** Ana Beatriz de Oliveira Monteiro, Lia Rosana Honnef, Júlia Meller Dias de Oliveira, Patrícia Pauletto, Graziela De Luca Canto.

**Investigation:** Ana Beatriz de Oliveira Monteiro, Lia Rosana Honnef, Júlia Meller Dias de Oliveira, Patrícia Pauletto, Carla Massignan, Cristine Miron Stefani, Glaucia Santos Zimmermann.

**Methodology:** Ana Beatriz de Oliveira Monteiro, Lia Rosana Honnef, Júlia Meller Dias de Oliveira, Patrícia Pauletto, Carla Massignan, Cristine Miron Stefani, Glaucia Santos Zimmermann, Graziela De Luca Canto.

**Supervision:** Lia Rosana Honnef, Júlia Meller Dias de Oliveira, Patrícia Pauletto, Carla Massignan, Cristine Miron Stefani, Glaucia Santos Zimmermann, Graziela De Luca Canto.

**Validation:** Ana Beatriz de Oliveira Monteiro, Lia Rosana Honnef, Júlia Meller Dias de Oliveira, Patrícia Pauletto, Carla Massignan, Cristine Miron Stefani, Glaucia Santos Zimmermann, Graziela De Luca Canto.

**Visualization:** Ana Beatriz de Oliveira Monteiro, Lia Rosana Honnef, Júlia Meller Dias de Oliveira, Patrícia Pauletto, Carla Massignan, Glaucia Santos Zimmermann, Graziela De Luca Canto.

**Writing – original draft:** Ana Beatriz de Oliveira Monteiro, Graziela De Luca Canto.

**Writing – review & editing:** Ana Beatriz de Oliveira Monteiro, Lia Rosana Honnef, Júlia Meller Dias de Oliveira, Patrícia Pauletto, Carla Massignan, Cristine Miron Stefani, Glaucia Santos Zimmermann.

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
