## [Decision Letter · Decision Letter 0]

30 Sep 2024

PONE-D-24-31009Prevalence of periodontitis in adolescents: a systematic review protocolPLOS ONE

Dear Dr. Pauletto,

Thank you for submitting your manuscript to PLOS ONE. After careful consideration, we feel that it has merit but does not fully meet PLOS ONE’s publication criteria as it currently stands. Therefore, we invite you to submit a revised version of the manuscript that addresses the points raised during the review process. Please note I have double-checked your submitted manuscript, and have commented on your draft's version (please see below). Additionally, our external Reviewers have forwarded their comments. Please double check, and revise your draft accordingly, or respond adequately to our external Reviewer's recommendations. . 

We look forward to receiving your revised manuscript.

Kind regards,

Andrej M Kielbassa

Academic Editor

PLOS ONE

Journal Requirements:

Additional Editor Comments :

Please additionally follow the comments forwarded by external reviewer #2:

The protocol demonstrates originality; however, to be deemed acceptable, a thorough revision addressing the points below is necessary:

Background: The current manuscript lacks a comprehensive presentation of the relationship between adolescents and periodontitis. I recommend a more thorough exploration of this relationship.

Methodology, Results, and Discussion: There is significant confusion in the formatting of the Results and Discussion sections. I request that these sections be reformulated and systematically organized.

Reviewers' comments:

Reviewer's Responses to Questions

**Comments to the Author**

1. Does the manuscript provide a valid rationale for the proposed study, with clearly identified and justified research questions?

Reviewer #1: No

Reviewer #2: Yes

2. Is the protocol technically sound and planned in a manner that will lead to a meaningful outcome and allow testing the stated hypotheses?

Reviewer #1: No

Reviewer #2: Partly

3. Is the methodology feasible and described in sufficient detail to allow the work to be replicable?

Reviewer #1: No

Reviewer #2: Yes

4. Have the authors described where all data underlying the findings will be made available when the study is complete?

Reviewer #1: No

Reviewer #2: Yes

5. Is the manuscript presented in an intelligible fashion and written in standard English?

Reviewer #1: No

Reviewer #2: No

6. Review Comments to the Author

You may also provide optional suggestions and comments to authors that they might find helpful in planning their study.

Reviewer #1: Abstract

- Please provide as much information as possible. Note that the word maximum is 300.

- Do not provide meaningless phrases with this section.

- "(...) its worldwide prevalence has been defined to be around 60%, (...)." This number would seem too low, depending on the the respective investigations. Please double check and adapt carefully.

- "No language or time of publication restrictions will be applied." This would seem mandatory, indeed. However, why do you restrict your research to four databases only (EMBASE, LILACS, PubMed/MEDLINE and Web of Science)?

- "Two independent authors will select the studies in a two-phase process based on predefined eligibility criteria." This does not correspond to your Meths section; there, you have stated that "The search will be done by the first author (A.B.O.M)."

- Even if considering "(L.R.H.)" as second reviewer AND "(J.M.D.O.)" to resolve possible disagreements, there is a total of 8 (EIGHT!) Co-Authors, but the tasks of the latter would not seem clear. Please clarify.

- Same with the current draft. Please clarify the responsibilities of all 8 Co-Authors.

Intro

- "What is the worldwide prevalence of periodontitis in adolescents?" While the main question would seem interesting, please provide a sound rationale. From the clinical experience it would seem clear that periodontitis is not a daily issue with adolescents. Please elaborate both aims and objectives more clearly.

- "(...), being capable of causing discomfort in the gingival region, bone damage and early tooth loss. Clear!" Please double check and revise carefully. What do you want to say when referring to "Clear!"?

- "That is why this study aims to identify the prevalence of this disease in adolescents." Please note that "no available SRs estimating the worldwide prevalence of periodontitis in adolescents" would not seem convincing. Remember that one possible reason for the observed non-availability could be that no researchers did see any sound reasons up to now. Again, clearly and convincingly elaborate why this study would seem mandatory, and discuss.

- Do you have any null hypotheses? Remember that H0 must be deducible from the forgoing thoughts.

Meths

- "A preliminary search strategy was conducted (...)." This again would seem unclear. Please provide both aims and results of your "preliminary search".

- Please provide the date of your PROSPERO pre-registration.

- Please provide the date of your start of literature research.

- Do not use legal terms. Delete "TM", "®", and so on.

- With ALL materials and methodologies (including statistical software), please use general names with your text, followed by (brand name; manufacturer, city, ST[ate - abbreviated, if US], country) in parentheses. Stick to semicolon. Remember that reproducibility must be ensured.

- "Two independent reviewer’s authors (A.B.O.M. and L.R.H.) will select the studies in two phases." Again, this would seem to contradict "The search will be done by the first author (A.B.O.M)." Please clarify.

- Same with "and a second reviewer (L.R.H.) will crosscheck (data from the included studies)". This would mean that ONLY those studies found by A.B.O.M. will be crosschecked, right? What about the studies A.B.O.M. has missed/overlooked/ignored? Who will really control A.B.O.M.'s work?

Disc

- "To date, no SR has explored the prevalence of periodontitis in adolescents." Please see comments given above. This would not seem a convincing rationale. Please discuss carefully but thoroughly.

- Please note that this section has not been convincingly elaborated, and clearly would seem perfectible.

Contributions of authors

- See comments given above. Please provide the aspects of the CURRENT draft. There has been no Data curation, no Formal analysis, and no Investigation up to now.

Refs

- Please revise for uniform Journal style.

Due to several major aspects, the current draft would not seem acceptable. With your careful revision, please stick to the Guidelines for Authors at https://journals.plos.org/plosone/s/submission-guidelines#loc-systematic-reviews-and-meta-analyses, to https://www.prisma-statement.org/, and to the respective PDF templates given there. Provide enough methodological detail to make the study reproducible and replicable.

Reviewer #2: Dear authors,

I would like to thank you for submitting your article to the journal. After a careful analysis of the manuscript, I inform you that it was accepted with major corrections. I suggest that the full text be revised to correct any inadequacies in the sentences. The introduction should also be revised to establish connections between the sentences, to address periodontitis in children in greater detail, to reformulate the second paragraph to avoid making it too long, and to summarize the quantitative results.

Identified Errors:

• In line 77, the word “guardians” is repeated and requires correction.

• In line 79, the term “clear” is employed inappropriately; we advise its removal from this line.

I appreciate your interest in publishing this project in the journal and look forward to receiving the revised manuscript with the necessary modifications.

7. PLOS authors have the option to publish the peer review history of their article (what does this mean? ). If published, this will include your full peer review and any attached files.

**Do you want your identity to be public for this peer review?** For information about this choice, including consent withdrawal, please see our Privacy Policy .

Reviewer #1: No

Reviewer #2: No

---

## [Author Response · Author response to Decision Letter 1]

27 Oct 2024

Dear Dr. Emily Chenette

Editor-in-Chief

PLOS ONE

We appreciate the opportunity to revise our article. We have carefully worked on incorporating the reviewers' suggestions to enhance the quality of the manuscript. Text modifications and additions were highlighted in yellow.

Reviewer #1

1. “Please provide as much information as possible. Note that the word maximum is 300. Do not provide meaningless phrases with this section.”

Response: We sincerely appreciate the valuable feedback provided. Based on the suggestions received, we have worked carefully to improve the mentioned section and ensure that the manuscript meets the expected standards. We are confident that the revisions have strengthened the quality of the work. We have added important information in the abstract, considering the methods used and objectives of the research, ensuring that the essential content is presented clearly and objectively, according to the maximum number of words. The modifications were highlighted and they can be found on page 2 and 3, lines 37 to 52, as follows in red:

A search will be conducted in the following databases: EMBASE, LILACS, PubMed/MEDLINE, and Web of Science. The gray literature will also be consulted through Google Scholar and ProQuest. No language or time of publication restrictions will be applied. To determine which studies will be included, two independent reviewers will select the articles through a two-phase process. In the first phase, reviewers will assess the title and abstract, and in the second phase they will read the full text using predefined eligibility criteria. Data collected will include author, study design, year of publication, country, sample size, sex, age, diagnostic methods, definition of periodontitis (including by sex), reliability of criteria, examination protocol, case classification and prevalence rates.

The risk of bias will be assessed according to the JBI Critical Appraisal Checklist for Studies Reporting Prevalence Data. Results will be synthesized into a narrative description and, if possible, a meta-analysis will be conducted. The systematic review methodology will enable a comprehensive and accurate analysis of the available evidence. Moreover, this approach may contribute to clinical decision-making, promoting the use of more appropriate and preventive treatments. (Page 2 and 3)

2. "(...) its worldwide prevalence has been defined to be around 60%, (...)." This number would seem too low, depending on the the respective investigations. Please double check and adapt carefully.

Response: We appreciate your insightful comments. The information in question has been thoroughly reviewed. The modifications were highlighted and they can be found on page 2, lines 30 to 32, as follows in red:

Periodontitis is a common periodontal condition in the adult dentate population, affecting approximately 6 in 10 individuals. Among those affected, approximately 25% develop the most severe form of the disease. The prevalence in adolescents is not well established.

Reference: Trindade D, Carvalho R, Machado V, Chambrone L, Mendes JJ, Botelho J. Prevalence of periodontitis in dentate people between 2011 and 2020: A systematic review and meta-analysis of epidemiological studies. J Clin Periodontol. 2023;50(5):604-26.

3. "No language or time of publication restrictions will be applied." This would seem mandatory, indeed. However, why do you restrict your research to four databases only (EMBASE, LILACS, PubMed/MEDLINE and Web of Science)?”

Response: Thank you for the feedback. We have followed the orientations from the Cochrane Handbook for Systematic Reviews. As recommended by the Handbook, the search should be conducted in at least three databases. Additionally, gray literature will also be considered.

Reference: Lefebvre C, Glanville J, Briscoe S, Featherstone R, Littlewood A, Metzendorf M-I, Noel-Storr A, Paynter R, Rader T, Thomas J, Wieland LS. Chapter 4: Searching for and selecting studies [last updated September 2024]. In: Higgins JPT, Thomas J, Chandler J, Cumpston M, Li T, Page MJ, Welch VA (editors). Cochrane Handbook for Systematic Reviews of Interventions version 6.5. Cochrane, 2024. Available from www.training.cochrane.org/handbook.

4. "Two independent authors will select the studies in a two-phase process based on predefined eligibility criteria." This does not correspond to your Meths section; there, you have stated that "The search will be done by the first author (A.B.O.M)."

5. "Two independent reviewer’s authors (A.B.O.M. and L.R.H.) will select the studies in two phases." Again, this would seem to contradict "The search will be done by the first author (A.B.O.M)." Please clarify.

Response: We appreciate your feedback. One author will apply the search strategy to the databases. After that, two authors will select the studies in Rayyan software. We tried to clarify this in the text as follows:

The first author (A.B.O.M) will apply the search strategy in each database. (Line 150 and 151, page 7). In the screening software, two independent reviewer’s authors (A.B.O.M. and L.R.H.) will select the studies in two phases. In phase 1, they will evaluate the titles and abstracts according to the eligibility criteria. After that, in phase 2, they will analyze the full-texts and select articles by the same criteria. Also, they will crosscheck all the results found. If disagreements arise, a third reviewer (J.M.D.O.) will make a final decision on both phases. (Line 159 to 163, page 7 and 8).

6. Even if considering "(L.R.H.)" as second reviewer AND "(J.M.D.O.)" to resolve possible disagreements, there is a total of 8 (EIGHT!) Co-Authors, but the tasks of the latter would not seem clear. Please clarify.

7. Same with the current draft. Please clarify the responsibilities of all 8 Co-Authors.

Response: Thank you for your comment. The authors of this protocol will be the same ones who will develop the systematic review. Each author will play an important role in developing the systematic review. In order to carry out a proper review, a group of experts is needed, both in the research topic and the methodological area. We've added everyone's contribution to the manuscript. We emphasize that all contributions followed the guidelines of the International Committee of Medical Journal Editors (ICMJE), ensuring fair attribution of authorship.

This systematic review protocol was produced, reviewed, and approved by all authors.

A.B.O.M. will work as a first reviewer on study conceptualization, study selection, data collection, data analysis and the initial manuscript draft.

L.R.H. and J.M.D.O. will work as the second and third reviewers, respectively. They will participate in study conceptualization, study selection, data collection, data analysis, and revising the initial manuscript draft.

P.P., C.M.S., G.Z.S., and C.M. will work on study conceptualization, study design, and data analysis. They will review the draft and approve the final manuscript. They compose a team of experts. C.M.S. and G.Z.S. are Periodontists, P.P. is Implantologist, and C.M. background is in Pediatric Dentistry. All are experienced with systematic reviews.

G.D.L.C. will work on study conceptualization, study design, data analysis, draft review and final manuscript approval before submission. She is the team's supervisor and an expert in systematic review methodology. (Line 280 to 293, page 12 and 13)

8. "What is the worldwide prevalence of periodontitis in adolescents?" While the main question would seem interesting, please provide a sound rationale. From the clinical experience it would seem clear that periodontitis is not a daily issue with adolescents. Please elaborate both aims and objectives more clearly.

Response: Thank you for the feedback received. Based on the valuable observations, we have revised and improved the original text, focusing on the clarity of the objectives and the strengthening of the rationale. The document now presents a more robust rationale, considering clinical experience as well as the risks associated with periodontitis in adolescents, such as early tooth loss and its impact on systemic diseases. The modifications will enhance the content's clarity and suitability regarding the proposed topic, as seen below.

A SR on this topic is important not only for clarifying the overall prevalence of periodontitis among adolescents but also for identifying high-risk groups through the identification of associated factors. This knowledge makes it possible to concentrate efforts on preventing and treating periodontitis for those in most need, which can minimize the progression of the disease in adulthood. Finally, for its global relevance since there are regional and socioeconomic variations in the impact of periodontitis on adolescents and studies on prevalence in different populations can offer valuable insights for public health. (Page 4 and 5, line 93 to 100).

Reference: Hung M, Kelly R, Mohajeri A, Reese L, Badawi S, Frost C, et al. Factors Associated with Periodontitis in Younger Individuals: A Scoping Review. J Clin Med. 2023;12(20).

9. "(...), being capable of causing discomfort in the gingival region, bone damage and early tooth loss. Clear!" Please double check and revise carefully. What do you want to say when referring to "Clear!"?

Response: We appreciate the valuable contributions that enabled this revision, ensuring that the content meets the required standards. Following a careful analysis of the text and considering the received suggestion, we inform you that the term "clear!" previously mentioned has been removed.

10. "That is why this study aims to identify the prevalence of this disease in adolescents." Please note that "no available SRs estimating the worldwide prevalence of periodontitis in adolescents" would not seem convincing. Remember that one possible reason for the observed non-availability could be that no researchers did see any sound reasons up to now. Again, clearly and convincingly elaborate why this study would seem mandatory, and discuss.

11. "To date, no SR has explored the prevalence of periodontitis in adolescents." Please see comments given above. This would not seem a convincing rationale. Please discuss carefully but thoroughly.

Response: We sincerely appreciate the feedback received. Based on the valuable contributions, I have modified the text according to the response presented to question 8 above (modifications in the text may be found on page 4 and 5, line 93 to 100, highlighted in yellow). We hope the modification will enhance the clarity and reinforce the importance of the topic.

Reference: Associated with Periodontitis in Younger Individuals: A Scoping Review. J Clin Med. 2023;12(20).

12. Do you have any null hypotheses? Remember that H0 must be deducible from the forgoing thoughts.

Response: We appreciate your feedback. In a systematic review, the use of a null hypothesis (or even a formal hypothesis) is not as common as in empirical or experimental studies. This is because the main goal of a systematic review is to synthesize and critically analyze the existing literature to answer a given research question rather than directly testing a hypothesis through the collection of new data. Besides, since this is a systematic review of prevalence, it is expected to present a more descriptive nature, in contrast to systematic reviews of intervention, to which hypothesis test is expected.

Reference: Cumpston M, Lasserson T, Flemyng E, Page MJ. Capítulo III: Relatando a revisão [última atualização em agosto de 2023]. Em: Higgins JPT, Thomas J, Chandler J, Cumpston M, Li T, Page MJ, Welch VA (editores). Cochrane Handbook for Systematic Reviews of Interventions versão 6.5. Cochrane, 2024. Disponível em www.training.cochrane.org/handbook.

13. "A preliminary search strategy was conducted (...)." This again would seem unclear. Please provide both aims and results of your "preliminary search".

Response: The development of a preliminary search strategy with the assistance of Medical/healthcare librarians and information specialists allows the identification of reference articles on the topic. This endorses the feasibility of the systematic review and helps the review team improve the definitive search strategy. We have clarify it in the text as follows:

A preliminary search strategy was developed with the assistance of a medical/health librarian and information specialists. This search strategy will be adapted for the following databases: EMBASE, LILACS (in Spanish: Literatura Latinoamericana y del Caribe en Ciencias de la Salud), PubMed/MEDLINE, and Web of Science. (Line 143 to 145, page 7).

14. Please provide the date of your PROSPERO pre-registration.

Response: Thank you for your comment. We have added the number at the end of the abstract and in the methods section, as follows:

This SR protocol was developed following the guidelines of Preferred Reporting Items for Systematic Review and Meta-Analyses Protocols (PRISMA-P) (Appendix 1), and it was registered in the International Prospective Register of Systematic Reviews (PROSPERO) on July 17, 2024, database under number CRD42024566374. (Line 108 to 109, page 5).

15. Please provide the date of your start of literature research.

Response: This is a protocol for a systematic review. We have already applied the search on January 4, 2024

16. Do not use legal terms. Delete "TM", "®", and so on.

17. “With ALL materials and methodologies (including statistical software), please use general names with your text, followed by (brand name; manufacturer, city, ST[ate - abbreviated, if US], country) in parentheses. Stick to semicolon. Remember that reproducibility must be ensured.”

Response: We appreciate your comments. We apologize; after reviewing and considering your observations, we have made the necessary corrections.

The file from each database will be imported into a reference manager (EndNote-Web; Clarivate, Saint Helier, Jersey) where duplicates will be removed. Subsequently, the references will be imported into a screening software (Rayyan, Qatar Computing Research Institute, Data Analytics, Doha, Qatar). (Line 154 to 157, page 5).

The data will be stored in a data extraction form that will be developed using a spreadsheet program (Microsoft Excel; Microsoft Corporation, Redmond, WA, USA). (Line 169 to 170, page 8).

If quantitative synthesis is appropriate, a proportion and a subgroup meta-analysis will be performed using the statistical computing software (R Statistics version 3.4.4; The R Foundation for Statistical Computing, Vienna, Austria) using the random effect measure and considering the range 95% confidence level from each study was included. (Line 194 to 195, page 9).

18. Same with "and a second reviewer (L.R.H.) will crosscheck (data from the included studies)". This would mean that ONLY those studies found by A.B.O.M. will be crosschecked, right? What about the studies A.B.O.M. has missed/overlooked/ignored? Who will really control A.B.O.M.'s work?.

Response: The studies will be selected by two authors to reduce the bias. After the decision about inclusion, the data from these included studies will be extracted by A.B.O.M. and cross-checked by LRH. Yet, according to the Cochrane Handbook for Systematic Reviews, the selection of studies by one author is not forbidden.

“Ideally, screening of titles and abstracts to remove irrelevant reports should also be done in duplicate by two people working independently (although it is acceptable that this initial screening of titles and abstracts is undertaken by only one person).”

Reference: Lefebvre C, Glanville J, Briscoe S, Featherstone R, Littlewood A, Metzendorf M-I, Noel-Storr A, Paynter R, Rader T, Thomas J, Wieland LS. Chapter 4: Searching for and selecting studies [last updated September 2024]. In: Higgins JPT, Thomas J, Chandler J, Cumpston M, Li T, Page MJ, Welch VA (editors). Cochrane Handbook for Systematic Reviews of Interventions version 6.5. Cochrane, 2024. Available from www.training.cochrane.org/handbook.

19. Please note that this section has not been convincingly elaborated, and clearly would seem perfectible

Response: Dear Editor, we sincerely appreciate your observations. Following a thorough reading and analysis, and taking your suggestions

---

## [Decision Letter · Decision Letter 1]

29 Dec 2024

PONE-D-24-31009R1Prevalence of periodontitis in adolescents: a systematic review protocolPLOS ONE

Dear Dr. Pauletto,

Thank you for submitting your manuscript to PLOS ONE. After careful consideration, we feel that it has merit but does not fully meet PLOS ONE’s publication criteria as it currently stands. Therefore, we invite you to submit a revised version of the manuscript that addresses the points raised during the review process. Having intensively double checked your re-submitted draft, both our external reviewers have forwarded recommendations considered contrasting to some extent. Consequently, I also have inspected your revised submission (see R #1), to come to a more balanced decision. Please note that your manuscript would not seem satisfying, and is not considered ready to proceed. Indeed, some critical aspects would seem in need of a thorough discussion. With your re-revision, you should follow the reviewers' comments added below, to finalize your paper convincingly, and to meet both Plos One's quality standards and our readership's expectations. Please remember that one revision is considered standard.

We look forward to receiving your revised manuscript.

Kind regards,

Andrej M Kielbassa

Academic Editor

PLOS ONE

Reviewers' comments:

Reviewer's Responses to Questions

**Comments to the Author**

1. Does the manuscript provide a valid rationale for the proposed study, with clearly identified and justified research questions?

Reviewer #1: Yes

Reviewer #3: Yes

Reviewer #4: Partly

2. Is the protocol technically sound and planned in a manner that will lead to a meaningful outcome and allow testing the stated hypotheses?

Reviewer #1: No

Reviewer #3: Yes

Reviewer #4: Partly

3. Is the methodology feasible and described in sufficient detail to allow the work to be replicable?

Reviewer #1: No

Reviewer #3: Yes

Reviewer #4: Yes

4. Have the authors described where all data underlying the findings will be made available when the study is complete?

Reviewer #1: No

Reviewer #3: Yes

Reviewer #4: Yes

5. Is the manuscript presented in an intelligible fashion and written in standard English?

Reviewer #1: Yes

Reviewer #3: Yes

Reviewer #4: Yes

6. Review Comments to the Author

You may also provide optional suggestions and comments to authors that they might find helpful in planning their study.

Reviewer #1: - With your reply/your revision, you have indicated that "(your study) was registered in the International Prospective Register of Systematic Reviews (PROSPERO) on July 17, 2024 (...)". At the same time, you state that "we have already applied the search on January 4, 2024". Please note that "prospective registration aims to reduce bias in the conduct and reporting of research and to increase transparency; in addition, prospective registration of systematic reviews is argued to help preventing unintended duplication, thereby reducing research waste". See more information at https://systematicreviewsjournal.biomedcentral.com/articles/10.1186/s13643-021-01877-1.

- You might wish to go to https://pmc.ncbi.nlm.nih.gov/articles/PMC10123738/ for more reasons and information.

- Same with https://pubmed.ncbi.nlm.nih.gov/31492940/. Why don't you provide a hard set of criteria on your decision when to include or exclude a possibly biased paper? You state that "observational cross-sectional studies" will be included, but you do not provide the inclusion criteria when it comes to the quality of the respective studies."Discrepancies will be resolved with the third examiner" would not seem convincing.

- With all these aspects in mind, please go to https://journals.plos.org/plosone/s/submission-guidelines#loc-systematic-reviews-and-meta-analyses. Remember that your study must follow the guidelines given with "Systematic reviews and meta-analyses".

Reviewer #3: Thank you for the opportunity to review this paper, and particularly the additional revisions. This paper will provide important clarity for the adolescent population, who are often assumed to not have periodontis and may benefit from additional prevention and screening.

Reviewer #4: The authors have presented a study protocol for a systematic review on the prevalence of periodontitis among adolescents. The study protocol is interesting but has the following concerns:

1. It is unclear what this study protocol adds to the literature which cannot be comprehensively included in a Prospero registration.

2. The authors state there are no systematic reviews with a similar scope. I would urge the authors to do a more comprehensive search. Articles such as: Catunda RQ, Levin L, Kornerup I, Gibson MP. Prevalence of Periodontitis in Young Populations: A Systematic Review. Oral Health Prev Dent. 2019;17(3):195-202. has not been cited nor discussed.

3. It is unclear how aggressive periodontitis and periodontitis will be analyzed/ segregated since majority of pre 2017 studies focussed on aggressive periodontitis as a separate and in many cases sole entity in adolescents. The authors do mention that they would segregate based on the definition but it maybe insufficient to fully capture the complexity by just definition.

4. The authors cite the GBD paper on periodontitis in the background section. The GBD study only reports on severe periodontitis . This needs to be clarified to avoid misinterpretations.

5. At the end of the background section the authors discuss aiming to find high risk groups and associated risk factors. The rest of the methods don't support this aim.

7. PLOS authors have the option to publish the peer review history of their article (what does this mean? ). If published, this will include your full peer review and any attached files.

**Do you want your identity to be public for this peer review?** For information about this choice, including consent withdrawal, please see our Privacy Policy .

Reviewer #1: No

Reviewer #3: **Yes: ** Amanda Ross-White

Reviewer #4: No

---

## [Author Response · Author response to Decision Letter 2]

22 Jan 2025

Dear Dr. Emily Chenette

Editor-in-Chief

PLOS ONE

We appreciate the opportunity to revise our article. We have carefully worked on incorporating the reviewers' suggestions to enhance the quality of the manuscript. Text modifications and additions were highlighted in yellow.

Reviewer #1:

1. “With your reply/your revision, you have indicated that "(your study) was registered in the International Prospective Register of Systematic Reviews (PROSPERO) on July 17, 2024 (...)". At the same time, you state that "we have already applied the search on January 4, 2024". Please note that "prospective registration aims to reduce bias in the conduct and reporting of research and to increase transparency; in addition, prospective registration of systematic reviews is argued to help preventing unintended duplication, thereby reducing research waste". See more information at https://systematicreviewsjournal.biomedcentral.com/articles/10.1186/s13643-021-01877-1.You might wish to go to https://pmc.ncbi.nlm.nih.gov/articles/PMC10123738/ for more reasons and information. - Same with https://pubmed.ncbi.nlm.nih.gov/31492940/.”

Response: We appreciate the observation and acknowledge the importance of prospective registration of systematic reviews in enhancing transparency and preventing unintentional duplication, as highlighted. However, we would like to clarify that the initial search was a pilot test, it was conducted as a test to assess the feasibility of the study and refine the strings.

Subsequently, with the assistance of a librarian the final strategy for each database was constructed. Then, the protocol was registered with PROSPERO on July 17, 2024, ensuring that the subsequent stages of the study adhered to the standards of transparency and methodological rigor associated with registration. We remain committed to conducting research responsibly and aligning our processes with best practices whenever possible.

To clarify this, we have changed the text adding this information, as follows:

“The preliminary search strategy was developed and tested on PubMed on January 4, 2024. After the tests, with the assistance of a health sciences librarian and information specialists the final search strategy was developed for each database”.

(Page 7, line 151-153)

2. “Why don't you provide a hard set of criteria on your decision when to include or exclude a possibly biased paper? You state that "observational cross-sectional studies" will be included, but you do not provide the inclusion criteria when it comes to the quality of the respective studies.”

Response: We appreciate the observation and recognize the importance of applying rigorous criteria to evaluate the quality of included studies. As outlined in our protocol, we utilize established risk-of-bias assessment tools to identify and mitigate potential sources of bias in the selected studies. However, as recommended by Cochrane (“Therefore, review authors should systematically take into account risk of bias in results of included studies when interpreting the results of their review.”), we did not exclude the studies based on their quality.

Reference: Boutron I, Page MJ, Higgins JPT, Altman DG, Lundh A, Hróbjartsson A. Chapter 7: Considering bias and conflicts of interest among the included studies [last updated August 2022]. In: Higgins JPT, Thomas J, Chandler J, Cumpston M, Li T, Page MJ, Welch VA (editors). Cochrane Handbook for Systematic Reviews of Interventions version 6.5. Cochrane, 2024. Available from www.training.cochrane.org/handbook

Also, we intend to do a sensitivity analysis removing studies with a high risk of bias from the meta-analysis and analysing the impact of the decision to include them.

“A sensitivity analysis is a repeat of the primary analysis or meta-analysis in which alternative decisions or ranges of values are substituted for decisions that were arbitrary or unclear. For example, if the eligibility of some studies in the meta-analysis is dubious because they do not contain full details, sensitivity analysis may involve undertaking the meta-analysis twice: the first time including all studies and, second, including only those that are definitely known to be eligible.”

Reference: Deeks JJ, Higgins JPT, Altman DG, McKenzie JE, Veroniki AA (editors). Chapter 10: Chapter 10: Analysing data and undertaking meta-analyses [last updated November 2024]. In: Higgins JPT, Thomas J, Chandler J, Cumpston M, Li T, Page MJ, Welch VA (editors). Cochrane Handbook for Systematic Reviews of Interventions version 6.5. Cochrane, 2024. Available from www.training.cochrane.org/handbook.

3. "Discrepancies will be resolved with the third examiner" would not seem convincing. - With all these aspects in mind, please go to https://journals.plos.org/plosone/s/submission-guidelines#loc-systematic-reviews-and-meta-analyses. Remember that your study must follow the guidelines given with "Systematic reviews and meta-analyses".

Response: We appreciate the observation and recognize the importance of a robust and transparent approach to resolving discrepancies between reviewers during the study selection and data extraction processes.

We have revised the text as follows:

“In the absence of consensus among the reviewers (A.B.O.M and L.R.H.), the final decision regarding the risk classification of the studies, whether low or high, will be made by a third reviewer (J.M.D.O).” (Page 9, line 202 to 204).

We based our methodology in:

“Use (at least) two people working independently to determine whether each study meets the eligibility criteria. Ideally, screening of titles and abstracts to remove irrelevant reports should also be done in duplicate by two people working independently (although it is acceptable that this initial screening of titles and abstracts is undertaken by only one person). It is essential, however, that two people working independently are used to make a final determination as to whether each study considered possibly eligible after title/abstract screening meets the eligibility criteria based on the full text of the study report(s) (see MECIR Box 4.6.c).”

Reference: Lefebvre C, Glanville J, Briscoe S, Featherstone R, Littlewood A, Metzendorf M-I, Noel-Storr A, Paynter R, Rader T, Thomas J, Wieland LS. Chapter 4: Searching for and selecting studies [last updated September 2024]. In: Higgins JPT, Thomas J, Chandler J, Cumpston M, Li T, Page MJ, Welch VA (editors). Cochrane Handbook for Systematic Reviews of Interventions version 6.5. Cochrane, 2024. Available from www.training.cochrane.org/handbook.

Reviewer #4:

The authors have presented a study protocol for a systematic review on the prevalence of periodontitis among adolescents. The study protocol is interesting but has the following concerns:

1. “It is unclear what this study protocol adds to the literature which cannot be comprehensively included in a Prospero registration.”

Response: We appreciate the observation. While the PROSPERO registration provides an overview of the protocol, our study includes additional elements that justify the independent publication of the protocol. In particular, the protocol outlines the detailed methodology, inclusion and exclusion criteria, and risk-of-bias assessment procedures specific to our analysis, including a rigorous evaluation of cross-sectional observational studies. These aspects are important for the transparency and reproducibility of the research and are not fully addressed in a general PROSPERO registration.

Our protocol aims to contribute to the literature by providing a robust foundation for conducting the systematic review, detailing the specific aspects of our methodological approach and the criteria established to ensure the quality of the selected studies. Furthermore, the publication of a protocol makes it more accessible to the public, facilitating its retrieval, as seen, for instance, in searches on PubMed.

2. “The authors state there are no systematic reviews with a similar scope. I would urge the authors to do a more comprehensive search. Articles such as: Catunda RQ, Levin L, Kornerup I, Gibson MP. Prevalence of Periodontitis in Young Populations: A Systematic Review. Oral Health Prev Dent. 2019;17(3):195-202. has not been cited nor discussed.”

Response: We appreciate the suggestion and the recommendation of the article. We recognize the importance of conducting a comprehensive literature search. The study was identified subsequently, during the initial stages of the search. We appreciate the suggestion and inform that the article by Catunda et al. has been duly incorporated into the introduction.

CUnderstanding the prevalence of periodontitis in different populations is essential to guide prevention and treatment strategies, particularly in vulnerable groups, thereby helping to reduce the progression of the disease into adulthood. Given its global relevance and the regional and socioeconomic variations in the impact of periodontitis among adolescents, prevalence studies provide valuable data to support public health strategies [9]. The systematic review conducted by Catunda et al. (2019) investigated the prevalence of the disease in young individuals but presented significant methodological limitations, such as restricting studies to those published between 1999 and 2017, including case reports and case series, and the absence of a meta-analysis. To address these limitations, the present systematic review aims to estimate the prevalence of periodontitis in adolescents aged 13 to 18 years without restrictions regarding publication year, language, or country, focusing exclusively on cross-sectional studies. This study seeks to provide a precise and comprehensive estimate of this condition within this age group” (Page 4 and 5, line 95 to 108).

3. “It is unclear how aggressive periodontitis and periodontitis will be analyzed/ segregated since majority of pre 2017 studies focussed on aggressive periodontitis as a separate and in many cases sole entity in adolescents. The authors do mention that they would segregate based on the definition but it maybe insufficient to fully capture the complexity by just definition.”

Response: We appreciate the observation and acknowledge the complexity of differentiating aggressive periodontitis and periodontitis in adolescents, especially in studies prior to 2017. While we initially proposed segregation based on definitions, we agree that this approach may be insufficient to fully capture the complexity of these conditions.

To enhance the accuracy of our analysis, we intend to adopt a more detailed approach, considering the clinical characteristics and diagnostic criteria used in the studies, in addition to the formal definitions.

“Periodontitis should be diagnosed through clinical examination (alone or in combination with radiographic assessment), using clearly defined criteria for case definition, taking into account the clinical characteristics and diagnostic criteria applied in the studies, in addition to the formal definitions established.” (Page 6, line 129 to 124)

Reference: Heitz‐Mayfield, L.J. (2024). Conventional diagnostic criteria for periodontal diseases (plaque-induced gingivitis and periodontitis). Periodontology 2000.

4. “The authors cite the GBD paper on periodontitis in the background section. The GBD study only reports on severe periodontitis. This needs to be clarified to avoid misinterpretations.”

Response: We appreciate the observation. We acknowledge the importance of clarifying the focus of the GBD study. The introduction section has been revised accordingly.

“A study that collected data from 1990 to 2019, from the Global Burden of Disease 2019 study, reported that periodontitis was negatively associated with the level of socioeconomic development. Periodontitis was more frequent among those aged 55-59 years, but the incidence of periodontitis has shown an increasing trend among younger individuals.” (Page 3, line 68-72)

5. “At the end of the background section the authors discuss aiming to find high risk groups and associated risk factors. The rest of the methods don't support this aim.”

Response: We appreciate the thorough analysis. The methods section has been reviewed to ensure its full alignment with the study's objective, which has been appropriately revised.

The following presents the revised objective:

“Sensitivity analysis

A sensitivity analysis removing studies with a high risk of bias from the meta-analysis and analysing the impact of the decision to include them is planned. “

(Page 11, line 251-254)

---

## [Decision Letter · Decision Letter 2]

12 Feb 2025

PONE-D-24-31009R2Prevalence of periodontitis in adolescents: a systematic review protocolPLOS ONE

Dear Dr. Pauletto,

Thank you for submitting your manuscript to PLOS ONE. After careful consideration, we feel that it has merit but does not fully meet PLOS ONE’s publication criteria as it currently stands. Therefore, we invite you to submit a revised version of the manuscript that addresses the points raised during the review process.

Please note that the previous Reviewers did not find any time/have stepped back to re-review your re-submitted draft. Therefore, I have double-checked your revised manuscript, and have commented on your last version (please see comments referring to Reviewer #1 as given below). In total, your submitted manuscript would not seem acceptable, and the final decision would be depending on further revisions. Please note that one review is considered standard with Plos One, so carefully stick to the comments and recommendations, and remember that a further revision will not be possible.

We look forward to receiving your revised manuscript.

Kind regards,

Andrej M Kielbassa

Academic Editor

PLOS ONE

Journal Requirements:

Reviewers' comments:

Reviewer's Responses to Questions

**Comments to the Author**

1. Does the manuscript provide a valid rationale for the proposed study, with clearly identified and justified research questions?

Reviewer #1: Yes

2. Is the protocol technically sound and planned in a manner that will lead to a meaningful outcome and allow testing the stated hypotheses?

Reviewer #1: No

3. Is the methodology feasible and described in sufficient detail to allow the work to be replicable?

Reviewer #1: No

4. Have the authors described where all data underlying the findings will be made available when the study is complete?

Reviewer #1: No

5. Is the manuscript presented in an intelligible fashion and written in standard English?

Reviewer #1: Yes

6. Review Comments to the Author

You may also provide optional suggestions and comments to authors that they might find helpful in planning their study.

Reviewer #1: The Co-Authors have re-submitted a revised protocol, and most of the previous comments have been clarified. Some aspects, however, still are considered in need of revisions. Please note that it is important that users can distinguish high quality reviews.

Consequently, please explicitly stick to the following aspects:

- With reference to adequacy of the literature search, please provide reasons why you think that including EMBASE, LILACS , MEDLINE (via PubMed) and Web of Science. would seem sufficient. There are other reliable databases, but why should the latter be excluded?

- "One reviewer (A.B.O.M.) will collect data from the included studies, and a second reviewer (L.R.H.) will crosscheck them." What does "will crosscheck" mean? A simple and superficial crosscheck would not seem sufficient. Please note that best practice does require two review authors to INDEPENDENTLY determine (1) eligibility of studies for inclusion in systematic reviews and (2) data extraction. This involves checking the characteristics of a study against the elements of the research question.

- Same with data extraction. The latter must be performed in duplicate. Please clarify.

- Excluded and included studies must be separately considered. Excluded studies should be accounted for fully by the review authors, otherwise there is a risk that they remain invisible and the impact of their exclusion from the review is unknown.

- Please revise for typos, see "reviewer’s authors".

- With their paper, review authors must provide detail about research designs, study populations, interventions, comparators, and outcomes (including the sources of funding for the studies included in the review). The details should be sufficient for any appraisers to make a judgment about the extent to which the studies were appropriately chosen (in relation to the PICO). Please clarify.

- Same with RoB. Please assure to account for RoB in primary studies when interpreting/discussing the results of the review.

- Describe where all data underlying the findings will be made available when the study is complete.

- Ensure that your systematic review design provides an accurate and comprehensive summary of the results of the available studies that address the question of interest. Ensure reporting the key sequential steps in the conduct of a systematic review, to ensure future high quality assessment of your SR.

In total, please note that some aspects might be clear for the Co-Authors. However, clarification of the aspects given above should help with identification of a high quality systematic review, so please clearly stick to the comments given above.

7. PLOS authors have the option to publish the peer review history of their article (what does this mean? ). If published, this will include your full peer review and any attached files.

**Do you want your identity to be public for this peer review?** For information about this choice, including consent withdrawal, please see our Privacy Policy .

Reviewer #1: No

---

## [Author Response · Author response to Decision Letter 3]

13 Mar 2025

Dear Dr.

Editor-in-Chief

PLOS ONE

We appreciate the opportunity to revise our article. We have carefully worked on incorporating the reviewers' suggestions to enhance the quality of the manuscript. Text modifications and additions were highlighted in yellow.

Reviewer #1:

1. “With reference to adequacy of the literature search, please provide reasons why you think that including EMBASE, LILACS, MEDLINE (via PubMed), and Web of Science. would seem sufficient. There are other reliable databases, but why should the latter be excluded?”

Response: Thank you for the feedback. The selection was based on their comprehensive coverage of biomedical and health sciences literature, ensuring a broad and relevant retrieval of studies. We have followed the orientations from the Cochrane Handbook for Systematic Reviews. As recommended by the Handbook, the search should be conducted in at least three databases. Additionally, gray literature will also be considered.

Reference: Lefebvre C, Glanville J, Briscoe S, Featherstone R, Littlewood A, Metzendorf M-I, Noel-Storr A, Paynter R, Rader T, Thomas J, Wieland LS. Chapter 4: Searching for and selecting studies [last updated September 2024]. In: Higgins JPT, Thomas J, Chandler J, Cumpston M, Li T, Page MJ, Welch VA (editors). Cochrane Handbook for Systematic Reviews of Interventions version 6.5. Cochrane, 2024. Available from www.training.cochrane.org/handbook.

2. "One reviewer (A.B.O.M.) will collect data from the included studies, and a second reviewer (L.R.H.) will crosscheck them." What does "will crosscheck" mean? A simple and superficial crosscheck would not seem sufficient. Please note that best practice does require two review authors to INDEPENDENTLY determine (1) eligibility of studies for inclusion in systematic reviews and (2) data extraction. This involves checking the characteristics of a study against the elements of the research question.”

3. “Same with data extraction. The latter must be performed in duplicate. Please clarify.”

Response: We appreciate your feedback. However, according to the Cochrane Handbook for Systematic Reviews, data extraction by a single author is not prohibited, similar to the study selection process. We tried to clarify this in the text as follows:

One reviewer (A.B.O.M.) will extract data from the included studies, while a second reviewer (L.R.H.) will thoroughly verify all extracted information by comparing the data extracted with the original data in the included studies. (Line 184 to 186, page 8).

Reference: Li T, Higgins JPT, Deeks JJ. Chapter 5: Collecting data [last updated October 2019]. In: Higgins JPT, Thomas J, Chandler J, Cumpston M, Li T, Page MJ, Welch VA (editors). Cochrane Handbook for Systematic Reviews of Interventions version 6.5. Cochrane, 2024. Available from www.training.cochrane.org/handbook.

4. “Excluded and included studies must be separately considered. Excluded studies should be accounted for fully by the review authors, otherwise there is a risk that they remain invisible and the impact of their exclusion from the review is unknown.”

Response: We appreciate the valuable comments. We would like to emphasize that, as required by the Cochrane Handbook for Systematic Reviews, this step will be performed. All excluded studies and reasons for exclusion will be properly documented and reported, ensuring transparency and clarity regarding the impact of their exclusion on the analysis process. “A Cochrane Review includes a list of excluded studies called ‘Characteristics of excluded studies’, detailing the specific reason for exclusion for any studies that a reader might plausibly expect to see among the included studies. This covers all studies that may, on the surface, appear to meet the eligibility criteria but which, on further inspection, do not. It also covers those that do not meet all of the criteria but are well known and likely to be thought relevant by some readers. By listing such studies as excluded and giving the primary reason for exclusion, the review authors can show that consideration has been given to these studies.”

We will rigorously follow the Cochrane guidelines to ensure the review's integrity and quality. We tried to clarify this in the text as follows:

“If important data for the review are missing or unclear, an attempt will be made to contact the study's corresponding author. The excluded studies with reasons for exclusion will be presented in the Appendix 2 of the final review.” (Line 180 to 182, page 8)

Reference: Lefebvre C et al. Chapter 4: Searching for and selecting studies [last updated September 2024]. In: Higgins JPT, Thomas J, Chandler J, Cumpston M, Li T, Page MJ, Welch VA (editors). Cochrane Handbook for Systematic Reviews of Interventions version 6.5. Cochrane, 2024. Available from www.training.cochrane.org/handbook.

5. Please revise for typos, see "reviewer’s authors".

Response: Thank you for your comment. We revised the text accordingly.

6. “With their paper, review authors must provide detail about research designs, study populations, interventions, comparators, and outcomes (including the sources of funding for the studies included in the review). The details should be sufficient for any appraisers to make a judgment about the extent to which the studies were appropriately chosen (in relation to the PICO). Please clarify.”

Response: We appreciate your inquiry and recognize the significance of providing comprehensive details on the studies included in the review. This ensures transparency and facilitates a critical evaluation of the appropriateness of the selection process concerning the PICO framework. To enhance clarity, we have revised the text as follows:

“The data will be stored in a data extraction form that will be developed using a spreadsheet program (Microsoft Excel; Microsoft Corporation, Redmond, WA, USA). If disagreements arise, a third reviewer (J.M.D.O.) will be contacted for a final decision. The collected information will include the author, study design, publication year country, patient characteristics (sample size, sex, and age), clinical characteristics (diagnostic methods, definition of periodontitis cases, case of periodontitis by sex, periodontal criteria confidence, periodontal examination protocol, periodontitis/healthy or gingivitis and extension/distribution), and outcomes (prevalence rates).” (Line 189 and 194, page 9)

7. “Same with RoB. Please assure to account for RoB in primary studies when interpreting/discussing the results of the review.”

Response: We sincerely appreciate the thoughtful feedback provided. We will conduct a sensitivity analysis to assess the impact of bias risk by excluding studies with high risk of bias from the meta-analysis. This approach aims to evaluate the robustness of the results. In the discussion section, we will address the impact of studies with high risk of bias through this sensitivity analysis, which appropriately responds to the raised question. The revised paragraph is presented below:

“A sensitivity analysis removing studies with a high risk of bias from the meta-analysis and analysing the impact of the decision to include them is planned”. (Line 258 to 259, page 12)

8. “Describe where all data underlying the findings will be made available when the study is complete.”

Response: Thank you for your suggestion. We will use the Open Science Framework (OSF) platform as a repository to make the original data available, ensuring transparency and accessibility.

This will encompass adjustments to the inclusion and exclusion criteria, along with the justification for each change and an assessment of its impact on the review's results. We will use the Open Science Framework (OSF) platform as a repository to make the original data available, ensuring transparency and accessibility. (Line 164 to 166, page 8)

9. “Ensure that your systematic review design provides an accurate and comprehensive summary of the results of the available studies that address the question of interest. Ensure reporting the key sequential steps in the conduct of a systematic review, to ensure future high quality assessment of your SR.”

Response: We sincerely appreciate your valuable suggestion. We followed the PRISMA-P guidelines to conduct this protocol. The systematic review will be reported in accordance with the PRISMA 2020 recommendations.

---

## [Decision Letter · Decision Letter 3]

16 Mar 2025

Prevalence of periodontitis in adolescents: a systematic review protocol

PONE-D-24-31009R3

Dear Dr. Pauletto,

We’re pleased to inform you that your manuscript has been judged scientifically suitable for publication and will be formally accepted for publication once it meets all outstanding technical requirements.

Kind regards,

Andrej M Kielbassa

Academic Editor

PLOS ONE

Additional Editor Comments (optional):

Reviewers' comments:

Reviewer's Responses to Questions

**Comments to the Author**

1. Does the manuscript provide a valid rationale for the proposed study, with clearly identified and justified research questions?

Reviewer #1: Yes

2. Is the protocol technically sound and planned in a manner that will lead to a meaningful outcome and allow testing the stated hypotheses?

Reviewer #1: Yes

3. Is the methodology feasible and described in sufficient detail to allow the work to be replicable?

Reviewer #1: Yes

4. Have the authors described where all data underlying the findings will be made available when the study is complete?

Reviewer #1: Yes

5. Is the manuscript presented in an intelligible fashion and written in standard English?

Reviewer #1: Yes

6. Review Comments to the Author

You may also provide optional suggestions and comments to authors that they might find helpful in planning their study.

Reviewer #1: This re-submitted draft has been satisfyingly revised, and would seem ready to proceed now.

Please note that there have been updated recommendations meanwhile, and these do improve the Cochrane Handbook's guidelines. Notwithstanding, referring to the latter would seem acceptable, at least with reviews being done in 2025.

7. PLOS authors have the option to publish the peer review history of their article (what does this mean? ). If published, this will include your full peer review and any attached files.

**Do you want your identity to be public for this peer review?** For information about this choice, including consent withdrawal, please see our Privacy Policy .

Reviewer #1: No

---

## [Editor Report · Acceptance letter]

PONE-D-24-31009R3

PLOS ONE

Dear Dr. Pauletto,

I'm pleased to inform you that your manuscript has been deemed suitable for publication in PLOS ONE. Congratulations! Your manuscript is now being handed over to our production team.

Kind regards,

on behalf of

Prof. Dr. med. dent. Dr. h. c. Andrej M Kielbassa

Academic Editor

PLOS ONE